

# Assessing the tsunami mitigation effectiveness of a planned Banda Aceh Outer Ring Road (BORR), Indonesia

Syamsidik[1,4,], Tursina[1,3], Anawat Suppasri[2], Musa Al'ala[1,3], Mumtaz Luthfi[1], and Louise K. Comfort[5]

[1]Tsunami and Disaster Mitigation Research Center (TDMRC), Syiah Kuala University, Jl. Prof. Dr. Ibrahim Hasan, Gampong Pie, Banda Aceh 23233, INDONESIA

[2] International Research Institute of Disaster Science (IRIDeS), Tohoku University, Aramaki Aza-Aoba 468-1, Aoba-ku, Sendai 980-0845, JAPAN, email: suppasri@irides.tohoku.ac.jp

[3]Graduate student at Civil Engineering Department, Syiah Kuala University, Jl. Syeh Abdurrauf No.7, Banda Aceh 23111, INDONESIA

[4]Civil Engineering Department, Syiah Kuala University, Jl. Syeh Abdurrauf No. 7, Banda Aceh 23111, INDONESIA

[5]Graduate School of Public International Affairs, University of Pittsburgh, Pittsburgh, US, email: lkc@pitt.edu

*Correspondence to*: Syamsidik (syamsidik@tdmrc.org, syamsidik@unsyiah.ac.id)

**Abstract.** This research aimed to assess the tsunami flow velocity and height reduction produced by a planned elevated road spanned parallel to the coast of Banda Aceh called Banda Aceh Outer Ring Road (BORR). Cornell Multi-Grid Coupled Tsunami Model (COMCOT) was used to simulate eight scenarios of the tsunami. One of them was based on the 2004 Indian
Ocean tsunami. Two magnitudes of earthquake were used, that is, 8.5 and 9.15 Mw. Both the earthquakes were generated from the same source location as in the 2004 case, around the Andaman Sea. Land use data of the innermost layer of the simulation area were adopted based on the 2004 condition and the land use planning of the city for 2029. The results of this study reveal that the tsunami flow depths and flow velocities can be reduced by about 9% by using the elevated road for earthquake magnitude 9.15 Mw and about 22% for earthquake magnitude 8.5 Mw. Combined with the land use planning
2029, the elevated road could reduce the maximum flow velocities behind the road by about 72%. Notably, the proposed land use for 2029 will not be sufficient to deliver any effects on the tsunami mitigation without the elevated road structures. We recommend the city to construct the elevated road as this could be part of the co-benefit structures for tsunami mitigation. The proposed BORR appears to deliver significant reduction of impacts in the smaller intensity tsunamis compared to the 2004 Indian Ocean tsunami.

## 1 Introduction

Tsunami mitigations by means of structural measure are not always affordable in the case of developing countries. In contrast, the threats posed by tsunami are real and have the potential to deliver severe impacts on the coastal area and the
35 community at risk. Banda Aceh is one of the most severely affected cities due to the 2004 Indian Ocean tsunami; however, it is difficult to follow the guidelines demonstrated by the advanced countries using developing massive physical structures to mitigate the future impacts of tsunami. This is still beyond the financial capacity of the city. In contrast, based on the probabilistic tsunami hazard assessment, this area could potentially be affected by a tsunami larger than 0.5 m, that is, about 10% higher, annually (Horspool et al., 2014). Therefore, seeking alternative and economic ways to mitigate impacts of the
40 tsunami could help the city in creating a more resilient region. Modifying the morphology and land use of the coastal front of the area can reduce the tsunami wave energy (Ohta et al., 2013). The nonlinearity effects generated on the inland tsunami



wave run-up are closely related to the local topography of the area (Mori et al., 2017). The key parameters of reducing damages due to tsunami waves are decreasing the wave velocity and the inundation depths (Kreibich et al., 2009; Yamamoto et al., 2006). These are better represented by a quadratic Froude Number ($Fr^2$) (Ozer and Yalciner, 2011). Constructing a high seawall is costly. In the case of Banda Aceh, the estimated maximum tsunami height based on the 2004 Indian Ocean

tsunami was 15 m (Lavigne et al., 2009). Only the sea walls higher than 5 m could contribute to the destructive effects of tsunami as in the case of the 2011 Tohoku tsunami (Nateghi et al., 2016). The cost of the structure is unarguably expensive. Furthermore, the tsunami wave has a long wave characteristics, where blocking the wave will only indicate the delaying time of the wave to reach a certain area behind the seawall due to scour process (Chen et al., 2016). Inequalities of the hydrostatic forces generated around the seawalls and the process of overlapping will occur and destroy the structures (Ozer et al., 2015);

however, this could reduce the tsunami wave energy (Guler et al., 2018).

Another way to reduce the tsunami wave energy is by using an elevated road. The elevated road can be functioned as an inland tsunami defense structure that could stop the tsunami wave or reduce its intensity as revealed in the case of the 2011 Great East Japan Earthquake and Tsunami (GEJET) (Goto et al., 2012a). In the GEJET case, the Tobu Highway in Sendai was the maximum limit of the tsunami inundation area in Sendai of Miyagi Prefecture in Japan (Abe et al., 2012; Goto et al.,

2012b; Sugawara et al., 2012). A new 6-m elevated road is now being constructed in Sendai, for which this idea was adapted from the tsunami mitigation effects revealed by the Tobu Highway structure during the 2011 Tohoku tsunami (Suppasri et al., 2016). Japan is an exemplary nation that promotes multilevel tsunami mitigation measures, either by structural or nonstructural mitigation. From structural mitigation, the GEJET affected areas have been developing several massive structures to prevent future tsunami losses (Strusinska-Correira, 2017; Koshimura et al., 2014; Pakoksung et al., 2018).

As a result of its population and economic growth, Banda Aceh is planning to construct a road transect as a response to the traffic demands of the city. One of the most recently introduced plans is a road that will circle the city from its periphery. The proposed road is named as Banda Aceh Outer Ring Road (BORR). Initially, the road is only introduced by a road transect and the detail structure of the road is yet to be decided. This is part of a long-term development program as stipulated in its spatial planning that aims to regulate the city planning until 2029 (Government of Banda Aceh, 2009). In the

spatial planning, no new significant tsunami mitigation infrastructure has been included. The structural mitigation facilities that were developed between 2005 and 2010 include four escape buildings, one tsunami museum that also functioned as tsunami escape building, and several escape routes. A-7 km revetment structure was constructed between 2006 and 2010 to reshape the city's coastline and to prevent further coastal erosion problem (Syamsidik et al., 2015).

To address the gap as stated earlier, this research aimed to investigate the potential tsunami destructive impacts reduction

through an elevated road structure parallel to the coastline of Banda Aceh (BORR). Cornell Multi-Grid Coupled Tsunami Model (COMCOT), a two-dimensional horizontal model was utilized to numerically simulate the tsunami characteristics as well as to evaluate the reduction impacts of BORR. Two types of land use maps in 2004 and 2029 were used to evaluate the mitigation effect of future tsunamis. The evaluation of the performance of the elevated road to reduce the tsunami wave energy may contribute to a better city planning of Banda Aceh in a long-term development program.

## 2 The Study Area

Banda Aceh is situated at the northern part of Sumatra Island and is the largest city in the Aceh Province. Figure 1 presents the city location. The topography of the city is flat with no hilly region. The hilly region is located around 7 km outside the

40 city's borders. There are several coastal lagoons situated at the northern part of the city. The city was severely damaged by the 2004 Indian Ocean tsunami, which caused death of about 90,000 people (Doocy et al., 2007). Prior to the 2004 tsunami, no knowledge was available regarding the potential tsunami that resulted in zero prevention of the hazard. During the rehabilitation and reconstruction process led by Aceh-Nias Rehabilitation and Reconstruction Agency (*BRR Aceh-Nias*), the





city faced serious challenges in relocating its people to a safer area. This resulted in several houses to be built at the coastal area. Initially, it was proposed to desert about 500 m from its coastal line from any settlement and was aimed for coastal vegetation as a part of the tsunami mitigation or was named as green belt area. This was mentioned in the Master Plan for Rehabilitation and Reconstruction composed by Indonesia Development and Planning Agency (Bappenas Indonesia)

(BAPPENAS, 2005). The 14-year rehabilitation and reconstruction process until 2018 has failed to make it happen.

At present, the coastal population of the city is growing significantly due to return migration from the affected community and more affordable land prices/house rent fees in the coastal area compared to other places in the city (Syamsidik et al., 2017). Figure 1 presents the study area of this research. In Figure 1, several tsunami flow depths data of this city, published by NOAA, are presented in red dots (NOAA, 2018); by Tsuji et al. (2006), are indicated as blue dots; and in the forms of

tsunami poles, are represented by yellow triangles. These flow depths were later incorporated in the tsunami numerical results validation. Figure 2 presents the condition of the coastal area of Banda Aceh based on an aerial image captured by a drone in February 2018. There is a-7 km revetment structure constructed along the city coast to immediately recover the eroded coastline and to create a barrier between the sea and ponds. The revetment was completed in 2010. Later in 2015, the government constructed a road transect at the leeward of the revetment; however, since the revetment is often being

overtopped by waves, the road is frequently damaged by the waves. Figure 3 presents the revetment structure and the road behind the revetment.

A new spatial planning and regulation of the city was released in 2009. This was modified in 2012 to accommodate the tsunami reconstruction process and few ideas to mitigate the disaster impacts, such as tsunami; however, no concrete measure was included in the spatial planning document to structurally mitigate the tsunami impacts.

Under the revision process of the spatial planning in 2012, the Government of Banda Aceh set up a new plan to construct a road due to the traffic congestion in the city. The road was proposed to cater the mobility of the people from the periphery of the city and was named as Banda Aceh Outer Ring Road (BORR). Japan International Cooperation Agency (JICA) has once studied the project. At present, the road project is put on hold due to the increase in the land prices, but is still in the formal document of the city development. A series of discussions were conducted to include the tsunami mitigation measures in the

new planned road. There is an opportunity to modify the design of the road to an elevated road. Some alternatives were drawn. One of the most intense discussions was to elevate the road to 3 m from the initial ground; however, the impacts of the planned elevated road on tsunami wave energy are not clear. The BORR transect is presented in Figure 1. The BORR will pass some area of salt marshes where ponds existed as the major land use types before tsunami. After the tsunami, large area of the fishponds were damaged and were never been recovered. In the new spatial planning regulation of the city, the

area will be kept as it is and only minor changes are proposed.

### 3 Methods

#### 3.1 Tsunami numerical simulations

To measure the impacts of the tsunami waves on the city, two scenarios of the coastal morphology were considered, that is, (1) without BORR and (2) with BORR. Tsunami simulations were performed using the Cornell Multi-Grid Tsunami Coupled Model (COMCOT). The COMCOT is a hydrostatic model that uses leap-frog finite difference method to solve the shallow water equations (SWEs) with a staggered scheme. Both the nonlinear and linear shallow water equations can be

selected in the model. COMCOT is a two-dimensional horizontal model that calculates the depth-averaged velocities. The linear shallow water equations in spherical coordinate system used in COMCOT are as follows.

$$\frac{\partial \eta}{\partial t} + \frac{1}{R \cos \varphi} \left\{ \frac{\partial P}{\partial \psi} + \frac{\partial}{\partial \varphi} (\cos \varphi \, Q) \right\} = -\frac{\partial h}{\partial t}, \tag{1}$$





$$\frac{\partial P}{\partial t} + \frac{gh}{R\cos\varphi}\frac{\partial \eta}{\partial \psi} - fQ = 0, \tag{2}$$

$$\frac{\partial Q}{\partial t} + \frac{gh}{R}\frac{\partial \eta}{\partial \varphi} + fP = 0, \tag{3}$$

Meanwhile, for nonlinear shallow water equations, COMCOT applies the following equations.

$$\frac{\partial \eta}{\partial t} + \frac{1}{R\cos\emptyset}\left\{\frac{\partial P}{\partial \psi} + \frac{\partial}{\partial \emptyset}(\cos\emptyset Q)\right\} = -\frac{\partial h}{\partial t}, \tag{4}$$

$$\frac{\partial P}{\partial t} + \frac{1}{R\cos\emptyset}\frac{\partial}{\partial \psi}\left\{\frac{P^2}{H}\right\} + \frac{1}{R}\frac{\partial}{\partial \emptyset}\left\{\frac{PQ}{H}\right\} + \frac{gH}{R\cos\emptyset}\frac{\partial \eta}{\partial \psi} - fQ + F_x = 0, \tag{5}$$

$$\frac{\partial Q}{\partial t} + \frac{1}{R\cos\emptyset}\frac{\partial}{\partial \psi}\left\{\frac{PQ}{H}\right\} + \frac{1}{R}\frac{\partial}{\partial \emptyset}\left\{\frac{Q^2}{H}\right\} + \frac{gH}{R}\frac{\partial \eta}{\partial \emptyset} + fP + F_y = 0, \tag{6}$$

$$f = \Omega \sin\varphi, \tag{7}$$

$$F_x = \frac{gn^2}{H^{7/3}}P(P^2 + Q^2)^{1/2}, \tag{8}$$

$$F_y = \frac{gn^2}{H^{7/3}}Q(P^2 + Q^2)^{1/2}, \tag{9}$$

$$H = \eta + h \tag{10}$$

Here, $P$ is the volume fluxes in $x$-direction (east-west direction), which is equal to $hu$, and $Q$ is the volume fluxes in $y$-direction (south-north direction), which is equal to $hv$, where $h$ is the depth at the grid to the mean sea level, and $(u,v)$ are the velocities at $x$- and $y$-direction, respectively. Furthermore, $\eta$ is the water surface elevation, $(\varphi, \psi)$ are the latitude and longitude for spherical coordinate system, $R$ is the earth radius, g is gravitational acceleration, and $h$ is the water depth at the grid. The component of $-\partial h/\partial t$ denotes the effect of transient seafloor motion; the Coriolis force coefficient due to the earth's rotation is expressed as $f$. Meanwhile, $\Omega$ is for the rotation rate of the earth; $H$ is the total water depth. $F_x$ and $F_y$ represent the bottom friction in the $\psi$ and $\varphi$ direction, respectively; and $n$ is Manning's roughness coefficient. A complete explanation of the COMCOT module can be referred to Wang (2009).

### 3.2 Computational regions

We applied six layers of simulation domains, starting from Layer 1 that covers the largest numerical domain including the tsunami source around the Andaman Sea. The innermost layer was Layer 6 that encompasses the Banda Aceh city and has the smallest size of the grid. The nested grid system also allows us to include the nonlinear effects of the tsunami waves in the COMCOT simulation. Details of the grid specification are listed in Table 1. All layers in the simulation apply spherical coordinate system. Figure 4 presents the simulation layers applied in this study.

Bathymetry data for Layers 1–4 were adopted from the GEBCO data with resolution of 1 min for all scenarios. Meanwhile, for Layers 5 and 6, we used the bathymetry data measured by the Geospatial Information Agency of Indonesia for the case of tsunami 2004. For the scenarios of 2029, we used the bathymetry data measured by the Aceh Public Works Department measured in 2007. Topography data measured by the Japan International Cooperation Agency (JICA) in 2005 were used for land topography data. The data were later updated by the Banda Aceh Development and Planning Agency. For the elevated road, the topography data along the transect were altered to plus 3.0 m from the mean sea level. The elevations were considered affordable in terms of the construction cost for the city. The structure of the elevated road was assumed to sustain the tsunami wave forces. For these, no scouring or altered ground elevation were made due to the tsunami wave forces.

40



### 3.3 Earthquake scenarios

We used two magnitudes of the earthquake in the simulations, that is, magnitude 8.5 and 9.15 Mw. Based on the probabilistic tsunami hazards assessment, the magnitude 8.5 Mw could occur once in about 200–300 years (Sengara et al., 2008; Suppasri et al., 2012a, Suppasri et al., 2012b), or in another study, it was said to have an exceedence return period by 100 years (Burbidge et al., 2009). Here the 8.5 Mw earthquake has the focal depth of 10 km, with a displacement of 8.3 m, where the dip and slip angles were 8° and 110°, respectively. The magnitude 8.5 Mw was calculated as a single fault. Meanwhile, the multifault method was adopted for 9.15 Mw. The fault details of the 9.15 Mw followed Koshimura et al. (2009). Dimension of the rupture area was calculated using Wells and Coppersmith Formulae (Wells and Coppersmith, 1994). Deformation of the seafloor caused by the rupture area was calculated following the formulae suggested by Masingha and Smylie (1971) and Okada (1985). Initial sea surface level as results of earthquake generation are presented in Figure 5. The land use for Layer 6 was adopted based on two conditions, that is, (1) land use of the city in 2004 before the Indian Ocean tsunami and (2) land use of the city as in Banda Aceh Spatial Planning regulation for 2029. The impacts of the elevated road by imposing two scenarios of the road were then compared, that is, (1) with BORR and (2) without BORR.

### 3.4 Data and validation

There are 8 simulations in total as listed in Table 2. Validation of the simulation was done by comparing the Simulation #211 with the heights of the tsunami inundation in Banda Aceh as marked by several tsunami poles in the city (Sugimoto et al., 2010). We used the Aida functions to validate the numerical results (Aida, 1978) that are based on $K$ and $\kappa$ as follows.

$$\log K = \frac{1}{n}\sum_{i=1}^{n}\log K_i \qquad , \tag{11}$$

$$\log \kappa = \sqrt{\frac{1}{n}\sum_{i=1}^{n}(\log K_i)^2 - (\log K)^2} \qquad , \tag{12}$$

$$K_i = \frac{H_{obs-i}}{H_{sim-i}}, \tag{13}$$

where, $H_{obs-i}$ is the observed tsunami inundation height or depth at point $i$ and $H_{sim-I}$ is the tsunami inundation height or depth based on the simulation at point $i$. The value of $\kappa$ represents the variance of $K_i$. Meanwhile, $K$ represents the mean of $K_i$. Takeuchi et al. (2005) suggested that the model results are in good agreement if $0.8 \leq K \leq 1.2$ and $\kappa \leq 1.60$. Another study also suggests that if the value of $\kappa$ can be complied and the value of $K$ is slightly >1.05, the results can also be classified as "Good Enough" (Koshimura et al., 2009).

Variations in the land use were included by modifying the Manning roughness coefficients based on land cover of the area. Table 3 presents the values of the Manning coefficients included in the simulations as suggested by Li et al. (2012). Distribution of the manning coefficients used in the two types of land use, that is, the 2004 and 2029 land use, is presented in Figures 6 and 7, respectively.

## 4. Results

### 4.1. Validation of the 2004 Indian Ocean Tsunami

To validate the result, we used the 2004 Indian Ocean tsunami case with land use form adopted the situation before the tsunami (without BORR) or Simulation #211 as listed in Table 2. Validations of the initial wave forms and offshore tsunami



wave propagation have been done by several studies (Koshimura et al., 2009; Suppasri, 2011; Suppasri et al., 2010). The studies used the water level around a transect in the Andaman Sea captured by JASON 1 Satellite about 2 h after the 9.15 Mw earthquake on December 26, 2004. The agreement of the simulated offshore tsunami wave heights was found in good accordance by the two aforementioned studies. For the tsunami inundation heights and depths, the results of the validation

5     are presented in Table 4 using Aida parameters calculated based on the equations (11)–(13). Based on the results, we confirmed that the simulated reports are in accordance with the observed data provided by the NOAA data and tsunami poles in the city.

### 4.2 Impacts of the Elevated Roads

Using the two magnitudes of earthquakes to generate tsunami waves, the impacts of BORR were tested. The following section elucidates a series of comparisons of the maximum wave run-up in Banda Aceh.

### 4.2.1 Magnitude 8.5 Mw

Distribution of the tsunami flow depths caused by the 8.5 Mw of earthquake is presented in Figure 9. Due to the BORR structure, the area of the inundation could be reduced by about 22%. Table 5 provides comparisons for all the scenarios for tsunami inundation area. It is observed that the impacts of the land use changes are not significant to further reduce the tsunami inundation area. The 2029 land use, if combined with BORR, will only further reduce the tsunami inundation area

by about 1.2%. The BORR coupled with land use changes can reduce the inundation area that is deeper than 2 m by about 25%.

Figure 12 provides comparisons of the tsunami wave heights for the three transects that are relatively perpendicular to the coastline. At all transects, we could observe that the magnitude 8.5 Mw could still generate tsunami heights by about 3 m along the coastline. Tsunami could cover the BORR structure, in particular, at the area around transect B. Interestingly, the

tsunami inundation area behind the BORR structure at transect B is mostly located at the salt marsh area where no population resides. At the other transects the tsunami waves could be stopped by the BORR structure, provided that the structure can sustain the stability test produced by the waves. Without BORR (scenarios #111 and #112), the tsunami wave could reach about 2 km from the coastline as presented in transect A (Fig. 12). With BORR (scenarios #121 and #122), the tsunami run-up could be reduced to the area of about 0.8 km from the coastline (see transect B in Fig. 12). The area where

the bridges are located, the tsunami waves could travel about 6 km along the main rivers. Considering that the river embankment is higher than 1.5 m from the original soil surface, the tsunami wave along the river will be able to retain itself in the river's main channel. At present, the river embankment along this city is done at 3 m from the soil surface under several projects undertaken between 1989 and 1992.

This was also proven true in the case of dike impacts on reducing the tsunami wave heights during the 2011 Tohoku

earthquake and tsunami on the Ishinomaki city of Japan (Takagi and Bricker, 2014). Interestingly, the inland structures as represented by the elevated road managed to stop the tsunami inundation. This was possible as the elevated road could reduce the velocity of the tsunami wave.

### 4.2.2 Magnitude 9.15 Mw

Figure 10 presents the comparison on the maximum tsunami inundation depths based on the land use types as in the condition before the 2004 Indian Ocean tsunami without BORR to the condition with BORR for earthquake magnitude of 9.15 Mw. The comparison clearly indicates the changes made by the BORR in terms of tsunami inundation depths. In front



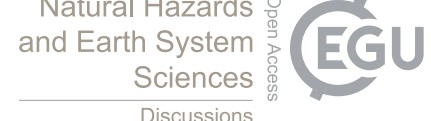

of BORR, the tsunami waves could be higher compared to the landward area of the road. In contrast, the tsunami inundation area could be 8.60% smaller if the road was constructed.

Similar effects of the BORR structures on the distribution of tsunami wave depths are presented in Figure 11 for the 2029 land use planning. Using the 2029 planned land use types with BORR, the wave heights could be decreased at the area

behind the road. In contrast, if we compare between Figure 10 (left) and Figure 11 (left), we would notice that the impacts of changing land use types as it is planned for 2029, will not have any significant difference in terms of the tsunami inundation depths and areas. Therefore, the changes of the land use alone are not sufficient to reduce the adverse impacts of the tsunami waves if the magnitude of the earthquake is 9.15. BORR coupled with the land use change (for 2029) could reduce the tsunami inundation area by about 9.7%.

All observed transects reveal similar effects of the BORR on the maximum inundation depths. The depths could be decreased after the BORR structure. Just at the leeside of the BORR structure, the depths will be decreased to about 5 m with the structure for earthquake magnitude of 9.15 Mw. This is about 28.5% lower than the situation without BORR. Figure 12 presents the comparison of maximum tsunami inundation depths for all the simulation scenarios for transects A, B, and C.

Maximum wave velocities at the area behind the proposed elevated road are listed in Table 6. The results proved that the

15 structure could significantly stop the tsunami in the case of earthquake magnitude of 8.5 Mw. For the earthquake magnitude of 9.15 Mw, the maximum velocities can be reduced by about 50% provided the land use is still the same as in 2004 and about 72% if the land use for 2029 is implemented. Herein, the modification of the land use combined with the BORR structures could potentially reduce the damages of the tsunami waves by about 22% lesser compared to the land use as in the 2004 case.

**5 Discussions and Limitations of the Study**

Effects of the elevated roads to limit the tsunami inundation demonstrated in the case of 2011 Great East Japan tsunami has inspired this research for Banda Aceh. This city was once severely damaged by the 2004 Indian Ocean tsunami. The inland structures and modification of the land use could help mitigate the impacts of tsunami waves. In our study, the proposed

elevated roads (BORR), planned to be constructed in Banda Aceh, which will be relatively parallel to the coastline, are expected to reduce the tsunami wave energy. This research found that the elevated road could effectively mitigate the tsunami generated by earthquake magnitudes of 8.5 and 9.15 Mw, generated around the Andaman Sea with different percentages of reduction. The larger the magnitude of the earthquake, less effective will be the reduction in the tsunami wave energy through BORR coupled with land use changes. As the land use is a dynamic variable, it is important to note that

certain land use controls to ensure the tsunami reduction effectiveness are necessary.

Based on the land use plan of Banda Aceh for 2029, the city will reclaim certain area around the coastal lagoons/salt marshes and will preserve some area of the lagoons as it is at present (salt marshes with mangrove forest). The lagoons play a significant role as they functioned as dug pools behind the revetment structures. In the case of overflow, the lagoons have the potential to reduce the tsunami wave energy as similar to that observed in the Teizan Canal of Tohoku area during the 2011

tsunami (Tokida and Tanimoto, 2014). The mangrove forests are also crucial in reducing the energy of tsunami waves as proven by several researches (see Yanagisawa et al., 2009; Iimura and Tanaka, 2012; Tanaka et al., 2014; Strusińska-Correia et al., 2013). In the case of inland embankment structure (such as the BORR structure in this study), the seaward coastal forest can reduce the possibility of overflow event. Furthermore, the landward forest could reduce the drag force behind the forest (Igarashi and Tanaka, 2018) and stop the tsunami debris. Therefore, it is important to preserve the area for the

mangrove forest and salt marshes.

Tsunami wave heights, as high as 3 m, can be reduced up to 1.5 m behind the structure of the elevated road, provided that the road structure is not breached. The concept of elevating the road to help mitigate impacts of the tsunami could be regarded as co-benefits development concept simultaneously integrating the traffic demands and tsunami mitigation. A

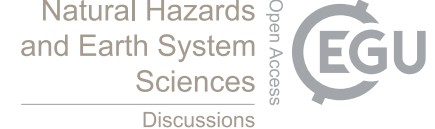



similar concept was observed in Sri Lanka to check the possibilities elevating train railway using embankment type of railway to reduce smaller intensity of tsunamis (Samarasekara et al., 2017). Adopting the principles of tsunami mitigation in the existing plan of the structure could also derive other impacts, such as the need to modify the city drainage system.

This study has certain limitations. Our proposed elevated road structure is an embankment type that has the elevation of plus 5.0 m from the mean sea level or around 3.5 m from the initial ground level. This type of structure will soon be covered by the tsunami waves if the magnitude of the earthquake is larger than 8.5 Mw. Since the waves are characterized as long-waves, scouring effects may occur immediately after the overlapping process. Furthermore, the leeside of the embankment will be easily damaged in the case of overlapping at a rubble mound type embankment (Aniel-Quiroga et al., 2018). The extreme difference on the hydrostatic pressures between the seaward and leeward direction of the BORR should also be considered. This could destabilize the structure (Ozer et al. 2015). This study excluded the damages that occurred due to the overflow process and scouring. Moreover, the density of the buildings was not considered as a parameter that could fluctuate the manning roughness coefficients as suggested by the previous researches (Kotani et al., 1998; Dutta et al., 2007).

**6 Conclusions and Recommendations**

This study explores the possibility to mitigate the impacts of future tsunami on Banda Aceh based on eight scenarios of numerical simulations. We used two magnitudes of earthquake that generate tsunami, that is, magnitudes 8.5 and 9.15 Mw. An elevated road and land use planning for 2029 were included in the simulations to test the possibility to adopt the concept of co-benefits structure for tsunami mitigation. Tsunami multidefense system as applied by Tohoku region after the 2011 Great East Japan Earthquake and Tsunami cannot be afforded for the tsunami prone cities in the developing countries, such as for Banda Aceh. There is a potential way to include the structural tsunami mitigation by modifying the coastal area profile. One of the possibilities for Banda Aceh is by elevating a planned road parallel to the coast, namely, Banda Aceh Outer Ring Road (BORR). Based on the simulations, the elevated road, by reclaiming plus 5.0 m from the mean sea level, could reduce the inundation area by about 9% and 22% in the case of 9.15 and 8.5 Mw of earthquake, respectively. The wave heights and the wave velocities could also be reduced using the elevated road structures. Notably, the land use planning alone without BORR will cause insignificant reduction in the tsunami wave heights and tsunami inundation area. Therefore, the elevated road coupled with the 2029 land use planning is expected to reduce the tsunami risks for the city, if implemented.

Based on the results, we recommend the Banda Aceh city to conduct several tsunami mitigation measures, as follows:

    a. to control the increase in population and settlements around the coastal area;

    b. to control the land use of the coastal area, in particular, the area in front of the planned BORR transect and to maintain it as a nonresidential area;

    c. to adopt the elevated roads in the BORR constructions as this will significantly help the city to cope with future tsunamis;

    d. to preserve the salt marshes area around the coast, as this would also help to reduce the tsunami impacts. The salt marshes area could also be planted with mangroves or other brackish water vegetation that would increase the manning roughness coefficients of the area. This further will reduce the speed of the tsunami waves.

**Acknowledgments**

The authors are grateful for the research grant from the Partnership Enhanced Engagement in Research (PEER) cycle 5 sponsored by the USAID and National Academies of Sciences of United States (NAS) under research grant #5-395, title "Incorporating climate change induced sea level rise information into coastal cities' preparedness toward coastal hazards"



with NAS Subaward No. 2000007546 This article was written under the World Class Professor Program (WCP) Scheme B, promoted by Ministry of Research, Technology, and Higher Education of Indonesia (RISTEKDIKTI) in 2018 (Contract No. No. 123.41/D2.3/KP/2018). Digitizing certain spatial data for land use and elevated roads was done under the PKLN of RISTEKDIKTI Program 2018.

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





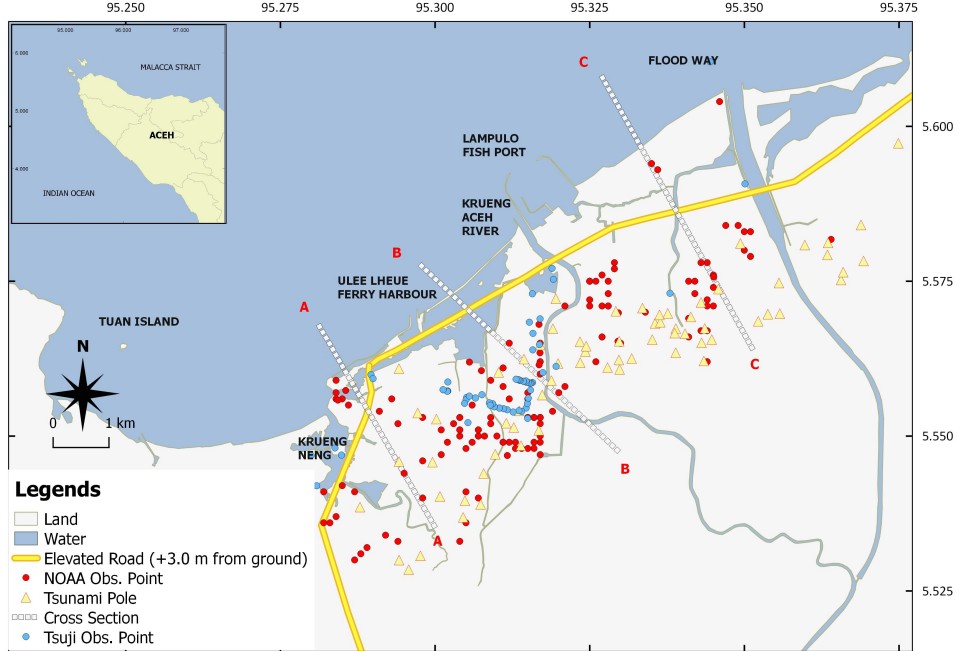

**Figure 1:** The study area. The elevated road (yellow line) is part of the city's development planning documents. There are 139 points of the 2004 tsunami heights measured by NOAA (red dots) (NOAA, 2018) and 56 locations of tsunami pole representing water marks based on eyewitness accounts (yellow triangles).



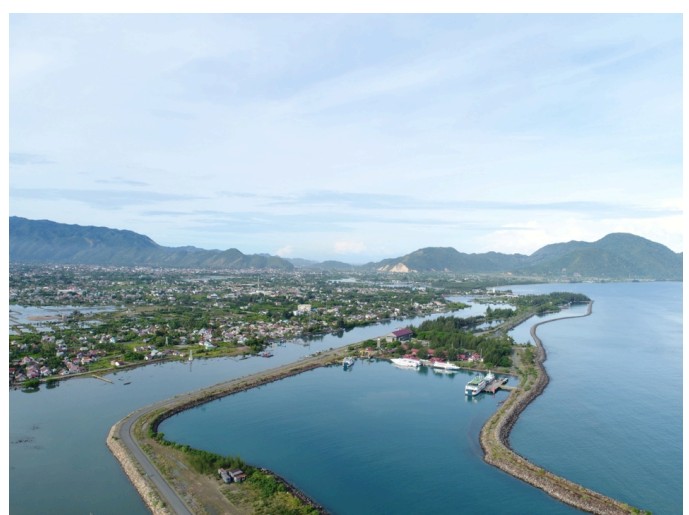

**Figure 2:** The situation of the coastal area of Banda Aceh based on aerial image taken in February 2018.





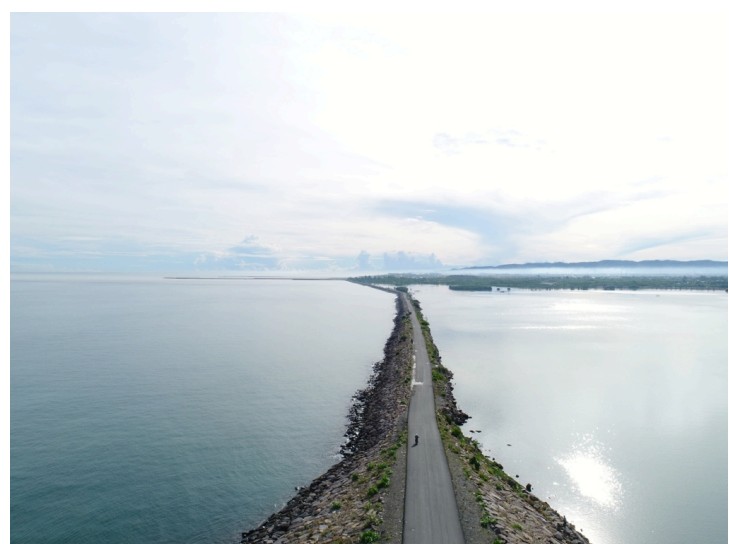

**Figure 3.** A 7-km embankment along the coast of Banda Aceh where a road was constructed at the leeward side of the embankment (Photo taken date, February 2018?).



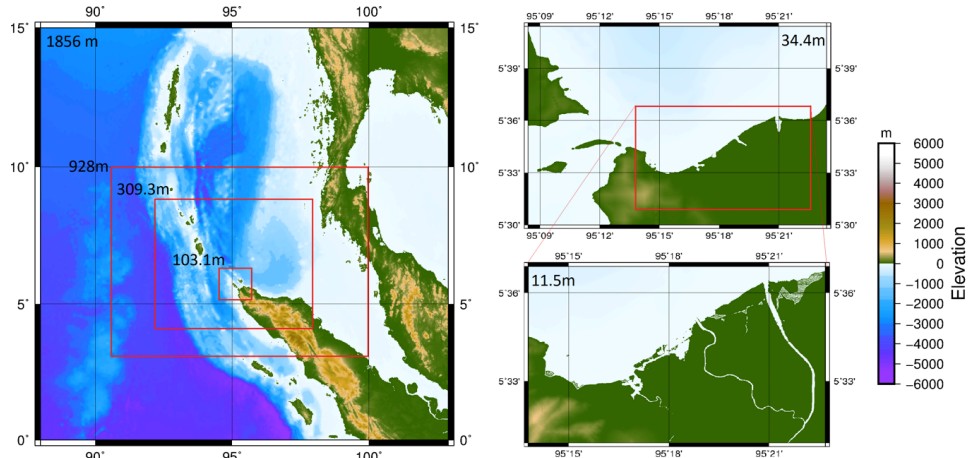

**Figure 4:** Six simulation layers and the size of the grids (written in each layer) applied in COMCOT.



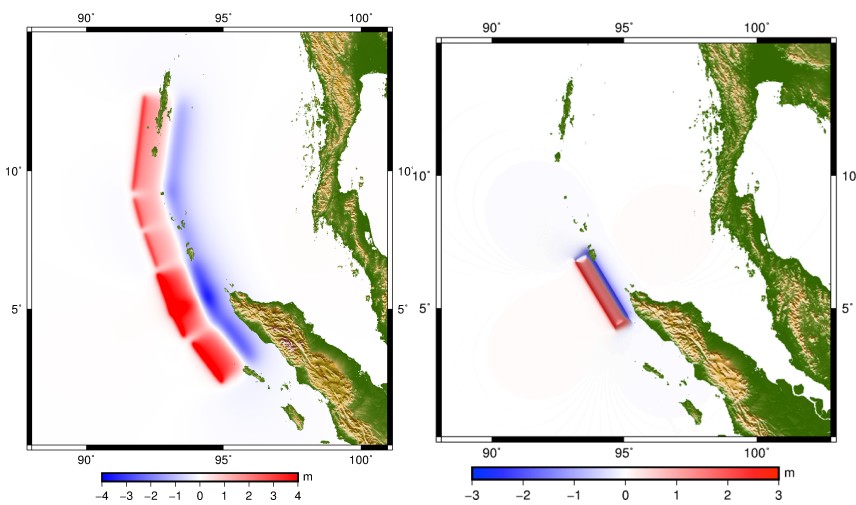

**Figure 5:** Initial wave forms of generated by the 9.15 Mw of earthquakes as proposed by Koshimura et al. (2009) (left) and by a hypothetical earthquake Magnitude 8.5 Mw (right).



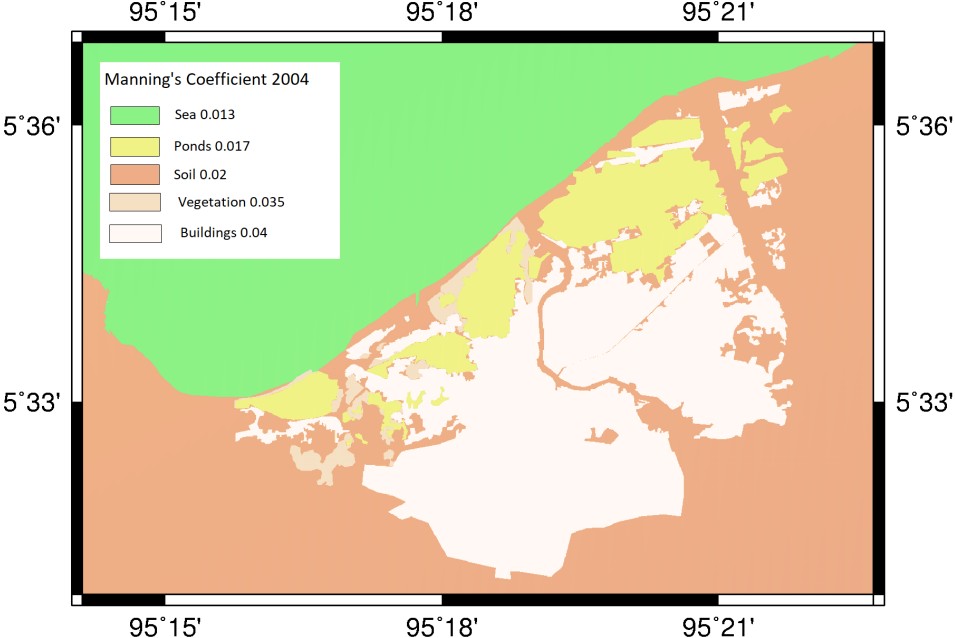

**Figure 6:** Distribution of manning coefficients used in the simulation for land use types in 2004 (before the 2004 Indian Ocean tsunami).





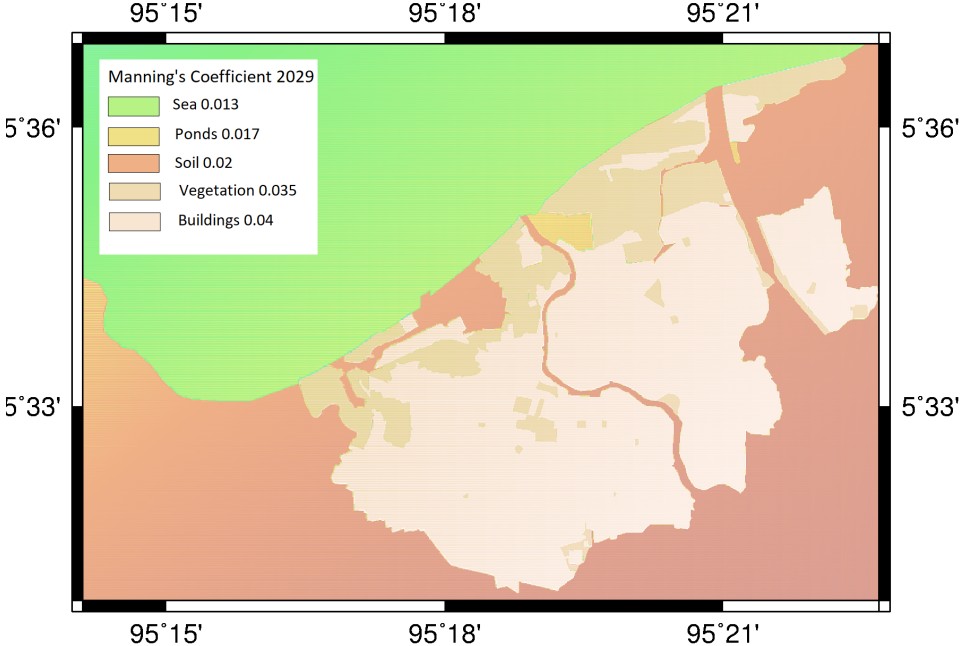

**Figure 7**: Distribution of manning coefficients used in simulations for land use types as described in the Banda Aceh spatial planning regulation aimed to be implemented until 2029.



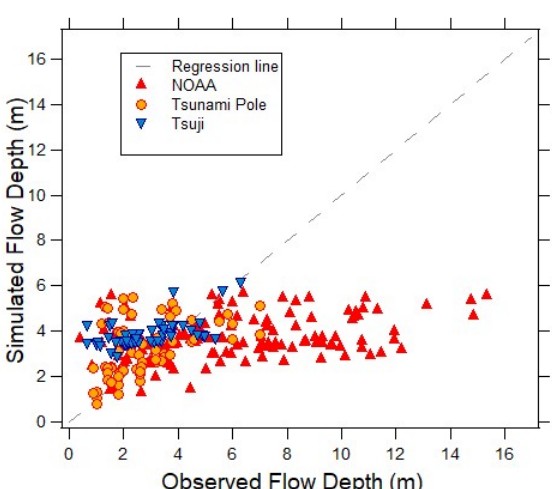

**Figure 8:**Comparisons between measured tsunami wave heights and simulation results.




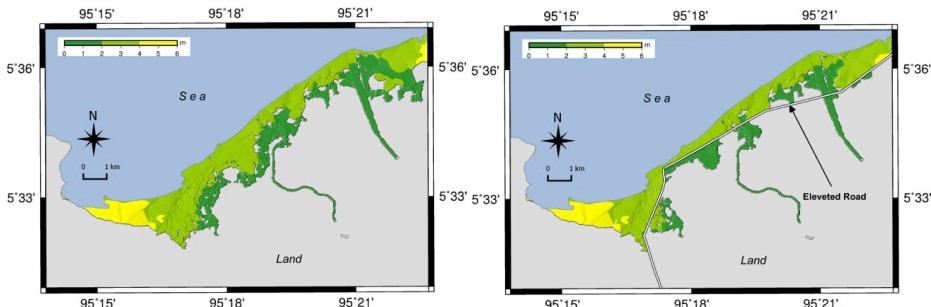

**Figure 9:** Comparison of maximum tsunami inundation depths generated by 8.5 Mw earthquake with condition without BORR (left) and with BORR (right).




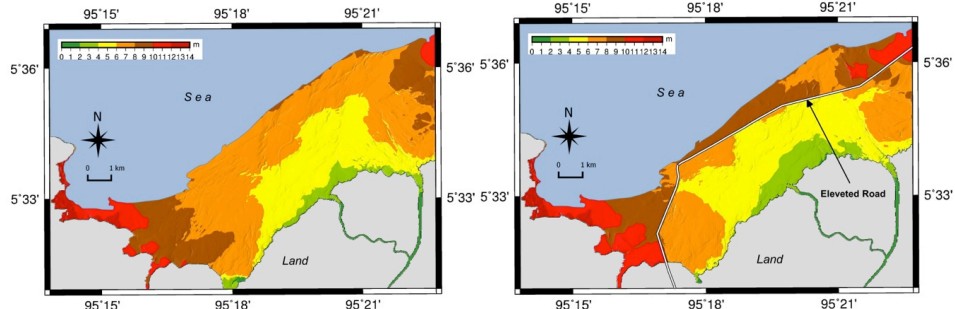

**Figure 10**: Maximum tsunami wave depths based on 9.15 Mw earthquake without BORR (left) and without BORR (right). The simulations were based on land use types before the 2004 Indian Ocean tsunami.





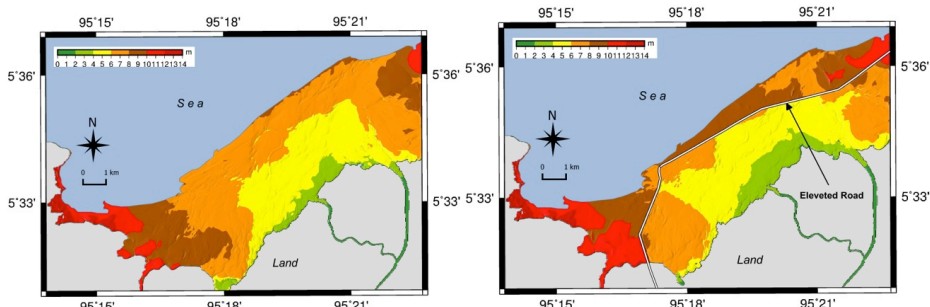

**Figure 11**: Maximum tsunami wave depths based on 9.15 Mw earthquake without BORR (left) and without BORR (right). The simulations were based on 2029 land use planning of Banda Aceh.





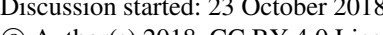

**Figure 12.** The comparison of tsunami wave heights at transects A, B, and C based on scenarios with and without BORR.





**TABLE 1**. Information on the setup of the five layer for COMCOT simulations.

| Layer | Latitude | Longitude | Number of Grid | Ratio | Grid size (m) | Time Step (sec.) | Manning Roughness Coefficients | SWE type |
|---|---|---|---|---|---|---|---|---|
| 1 | 0.1 / 14.93 | 88.1 / 102.8 | 1772 | 1 | 1856 | 0.2 | 0.013 | Linear |
| 2 | 3 / 10 | 91 / 100 | 1920 | 2 | 928 | 0.1 | 0.013 | Linear |
| 3 | 4.08 / 8.98 | 92.05 / 97.98 | 3899 | 3 | 309.33 | 0.03 | 0.013 | Linear |
| 4 | 5.2708 / 6.695 | 94.51 / 95.99 | 3137 | 3 | 103.11 | 0.011 | 0.013 | Linear |
| 5 | 5.5 / 5.69 | 95.14 / 95.39 | 1426 | 3 | 34.37 | 0.004 | 0.013 | Linear |
| 6 | 5.515 | 95.235 | 2362 | 3 | 11.5 | 0.001 | Varied Coefficients (see **Table 3**) | Nonlinear |



**Table 2.** Scenarios of the Numerical Simulations.

| Magnitude (Mw) | BORR Scenario | Land Use in Year | Code of Simulations |
|---|---|---|---|
| 8.5 | Without BORR | 2004 | #111 |
| | | 2029 | #112 |
| | With BORR | 2004 | #121 |
| | | 2029 | #122 |
| 9.15 | Without BORR | 2004 | #211 |
| | | 2029 | #212 |
| | With BORR | 2004 | #221 |
| | | 2029 | #222 |



**Table 3.** Manning Coefficients based on land cover of the area (Li et al., 2012)

| Land Use | Manning's Roughness Coefficient (n) |
|---|---|
| Coastal Vegetation | 0.035 |
| Fish Ponds | 0.017 |
| Building | 0.04 |
| Sea | 0.013 |
| Soil | 0.02 |



**Table 4**. The validation results of the simulation using Aida parameters for Simulation #211.

| Model Results | Aida parameters | |
|---|---|---|
| | *K* | *k* |
| NOAA Data (*n*= *139* ) | 1.18 | 1.42 |
| Tsunami Pole data (*n*= *56* ) | 0.79 | 1.50 |
| Tsuji et al., 2006 (*n*=50) | 0.68 | 1.61 |



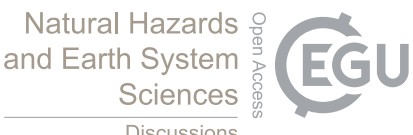

Table 5. Comparisons of tsunami inundation area based on the simulations.

| Magnitude of Earthquake | Land Use Type | Total area of Inundation (Ha) | Area of Inundation deeper than 2 m (Ha.) | % of total decrease | % of decrease for area deeper than 2 m |
|---|---|---|---|---|---|
| 8.5 Mw | 2004 without BORR | 1,591.73 | 998.89 | | |
| | 2004 with BORR | 1,252.20 | 746.38 | -21.33 | -25.28 |
| | 2029 without BORR | 1,553.03 | 979.58 | | |
| | 2029 with BORR | 1,203.47 | 741.67 | -22.51 | -24.29 |
| 9.15 Mw | 2004 without BORR | 4,654.27 | 3,722.60 | | |
| | 2004 with BORR | 4,254.17 | 2,121.17 | -8.60 | -43.02 |
| | 2029 without BORR | 4,592.60 | 3,561.26 | | |
| | 2029 with BORR | 4,148.91 | 1,991.13 | -9.66 | -44.09 |



**Table 6**. Maximum velocities after the elevated road structures.

| | Max. velocities for 8.5 Mw (m/s) | | | | Max. velocities for 9.15 Mw (m/s) | | | |
|---|---|---|---|---|---|---|---|---|
| | Without BORR | | With BORR | | Without BORR | | With BORR | |
| Transect | 2004 | 2029 | 2004 | 2029 | 2004 | 2029 | 2004 | 2029 |
| A | 4.52 | 3.80 | 0.00 | 0.00 | 4.85 | 4.98 | 2.85 | 1.40 |
| B | 3.20 | 1.30 | 0.25 | 0.22 | 4.42 | 5.15 | 1.90 | 1.51 |
| C | 0.45 | 0.41 | 0.00 | 0.00 | 4.92 | 4.60 | 2.35 | 1.25 |