# Peer review of "Assessing the tsunami mitigation effectiveness of a planned Banda Aceh Outer Ring Road (BORR), Indonesia"

_Natural Hazards and Earth System Sciences, 2018_

## Referee Comment (RC1) · Anonymous Referee #1 · 20 Nov 2018

"general comments"

This paper discussed the effectiveness of the elevated road as the construction for tsunami disaster mitigation in Banda Aceh by usuing numerical simulation . The elevated road was planned after 2004 Indian Ocean tsunami. However, the detailed evaluation of the road as the disaster prevention facility has not been conducted. In this paper, 4 scenarios were prepared for the tsunami inundation simulation. These scenarios included the change of land use in the city of Banda Aceh. Banda Aceh has been recovering and developing after 2004 tsunami disaster. Therefore, the viewpoint of land use is important to discuss the effect of the disaster prevention facility, such

as the elevated road, in near future. The scenerario of the magnitude of earhquake is quite severe, 8.5 and 9.15 Mw, and the reduction of the tsunami inundation area by the elevated road is not sufficient. But the effect of the road was confirmed and it is inferer that the road will have some effective functions for the disaster mitigation against the tsunami generated by earthquake smaller than 2004 Indian Ocean Tsunami. Consequently, it is expected that the results of this paper lead to the more detailed plannning of the elevated road and discussion as the disaster prevention facility. Then, this paper will contribute to the future tsunami disaster mitigation and development of Banda Aceh city.
* * *
"specific comments"

- In Abstract: The condition of BORR should be written breifly.

- p.1, L.38-39: The relationship between 0.5m and "10% higher" is unknown.

- p.3, L.2: "about 500m from its coastal line from any settlement". Where is it "500m" from?

- p.3, L.15-16: From Fig.3, is the road on the revetment?

- p.3, L.23: Has the problem (increase of land price) been solved already? Is there a possibility of restarting the project?

- p.4, L.13,14: In linear SWE, P=hu and Q=hv are correct. However, are these correct in nonlinear SWE?

- p.4, L.33-34: Where is the reference height for "3m". If it is from the mean sea level, the heigt of the elevated road is less than 3.0m. But if it is from the original ground level, the height is 3.0m. At p.3, L.26, the term of "initial ground" is used.

- p.4, L.33-34: How mucg is the width of the road? Is the grid size of 11.5m in Layer 5 enough for the description of the road?

- p.5, L.12-13: The land use in 2004 and its plan for 2029 should be shown. And the difference should be explained.

- p.6, L.19-20: How did you calculate the percentage of 1.2%? If this value is "% of total decrease" in Tab.5, what is the meaning of this value? If you want to say the effect of the land use change from 2004 to 2029 with BORR, you should calculate this value using 1252.0 (2004 with BORR) and 1203.47 (2029 with BORR). It is about 3.9% (>1.2%).

- p.6, L.24: What is the reason of "Interestingly"?

- p.6, L.29-30: Where are the bridges in Fig.12?

- p.6, L.31, "travel about 6km along the main rivers.": It is impossible to confirm the 6km-inundation in Fig.12.

- p.6, L.31: What is the reason of "higher than 1.5m"?

- p.6, L.36-37: There is no discussion about velocity before. Is it possible to mention the effect of velocity reduction by the elevated road here?

- p.7, L.9: It is hard to recognize this value (difference) in Fig.11. You should mention Tab.5 for this difference.

- p.7, L.11, "...about 5m with...": From the figure, "4m" is proper.

- p.7, L.18: What kind of "damages" do you considered? There is no explanation about the concrete type of damage.

- p.7, L.18: How did you calculate "about 22%"?

- p.7, L.29: What is the meaning of "dynamic variable"?

- p.7, L.41: From Fig.12, tsunami wave with 3m height does not overflow the structure. The content of this sentence is not consistent to the simulation results.

- p.8, L.3: What is the relation between the elevated road and the drainage system?

- p.8, L.4-5: Is this sentence consistent to "p.3, L.26" and Fig.12.

- p.8, L.19: What is the "co-benefits" for tsunami wave?

- p.8, L.19: What is "Tsunami multidefense system"? There is no explanation.
* * *
"technical corrections"(typing errors, etc.)

- p.4: In equation, "sin" and "cos" should be written in Roman style.

- p.4: "$\Phi$" should be changed to "$\varphi$" in eqs.(4)-(6).

- p.4, L.16: "$g$" should be written in Italic style.

- p.5, Eqs.(11) and (13): Are these descriptions correct? Is "logK" a variable, that is, is "log" not a function? Eq.(12) is the same.

- p.6, L.21: ... about 25% both in 2004 and 2029.

- p.6, L21: "Figure 12" should be labeled as "Figure 10". Because tis figure is refered before original Figure 10 and 11

- p.6, L.22: ... the three transect indicated in Fig.1 that ...

- p.6, L.29, "could be reduced": "stopped" may be suitable in this case than "reduced".

- p.6, L.41: The number of figures should be changed by change of figure number of Fig.12.

- p.7, L.35: "Tokida and Tanimoto, 2014" is not found in the references.

- p.8, L.7, "overlapping process": "overtopping" or "overflowing" ?

- Figure 4,5,9,10,11: These figures should be bigger.

- Figure 10 (cpation): The simulation were demonstrated by using land use ...

- Figure 11 (caption): ...and with BORR (right).

- Figure 12: What is "Elevated Roud ($\pm$5.0m)"? In p.4, L.34, "to plus 3.0m from the meas sea level"

- Figure 12: It is difficult to distinguish the difference of lines, especially yellow lines are unclear.

- Figure 12 (legend): Simulation code should be wriiten in the legend because the code is used in the main sentense.

- Table 1 (title): ...setup of the six layers for ...

- Table 1: What is the "Ratio" in 5th column? If this is grid size ratio from parent layer to child layer, a blank is better in Layer 1.

- Table 1 (Layer 6): Two values in Latitude and Longitude may indicate the locations of "start" and "end", respectively. But why is only one value in Layer 6?

- Table 5: What is "Ha"? Is this "ha" (hectare)?

- Table 5: Font size in the bottom row is slightly bigger than others.

---

## Referee Comment (RC2) · Anonymous Referee #2 · 5 Dec 2018

General comments: This paper assessed the tsunami mitigation effect of the future Banda Aceh Outer Ring Road (BORR) using numerical simulations. Considering the significant tsunami hazard that Banda Aceh has experienced during the 2004 Indian Ocean tsunami and the potential tsunami threat in the future, having tsunami mitigation structures is urgently important. The evaluation method is valid and the results are appropriate to support the general conclusion. Detailed comments: 1. For the earthquake scenarios, two magnitudes Mw 8.5 and Mw 9.15 are chosen. More justification is required to explain how the fault parameters (e.g. focal depth, dip and slip angle and slip value) are decided. For example, providing some evidences for the fault geometry. 2. Figure 10 and Figure 11, caption, correct to "...with BORR (right)" 3. Table 1. In

COMCOT, the Manning roughness coefficients will not function when the SWE type is "Linear", so the second last column should be set to "None" when the SWE type is "linear"

---

## Author Comment (AC1) · 8 Dec 2018

First of all, we thank to Referee #2 comments on our paper posted for discussion on December 5, 2018. We regard the comments with high appreciation and attempt to include them in our revised manuscript. The following sections are our responses to the comments. COMMENT 1: For the earthquake scenarios, two magnitudes Mw 8.5 and Mw 9.15 are chosen. More justification is required to explain how the fault parameters (e.g. focal depth, dip and slip angle and slip value) are decided. For example, providing some evidences for the fault geometry.

RESPONSE 1: Koshimura et al. (2009) proposed fault parameters for the 2004 Indian

[Figure]

Ocean tsunami case. The fault was divided into 6 segments where accumulative energy is similar to total energy generated by the fault. Details of the fault proposed by Koshimura et al. (2009) and their locations can bee seen in the Supplement part.

The result of this multi-fault has been validated at the onshore area of Banda Aceh using measured flow-depths and flow-heigths. More complete explanation of this can be seen in Koshimura et al. (2009). We decide not to include the table and the figure to allow readers to read a more complete and rigour studies done by Koshimura et al. as briefly explained here. For 8.5 Mw, we follow suggestions made by Horspool et al. (2014 ). We based our simulations on the parameters with strike of 329o, dip 8.0 o, slip 110o, and depth of 10 km. The 8.5 Mw simulation use single fault scenario where the location of the fault has been moved along the fault lines to obtain maximum impacts on Banda Aceh coast. We agree with the referee to add the explanation of the 8.5 Mw simulation fault scenario in or revised manuscript. Please see section 3.3 Earthquake scenarios in revised manuscript.

COMMENT 2: Figure 10 and Figure 11, caption, correct to ". . .with BORR (right)"

RESPONSE 2: Thank you for the detail correction. We will revise the caption on Figure 10 and Figure 11 ..... "without BORR (right)" with "with BORR (right)"

COMMENT 3: Table 1. In COMCOT, the Manning roughness coefficients will not function when the SWE type is "Linear", so the second last column should be set to "None" when the SWE type is "linear"

RESPONSE 3: Thank you very much. We will replace Manning Roughness Coefficients "0.02" to "None" as suggested. The revised table can be seen in the Supplement part of these responses.

Please also note the supplement to this comment:
https://www.nat-hazards-earth-syst-sci-discuss.net/nhess-2018-276/nhess-2018-276-AC1-supplement.pdf

**Supplement:**

First of all, we thank to Referee #2 comments on our paper posted for discussion on December 5, 2018. We regard the comments with high appreciation and attempt to include them in our revised manuscript. The following sections are our responses to the comments.

**COMMENT 1:**
For the earthquake scenarios, two magnitudes Mw 8.5 and Mw 9.15 are chosen. More justification is required to explain how the fault parameters (e.g. focal depth, dip and slip angle and slip value) are decided. For example, providing some evidences for the fault geometry.

**RESPONSE 1:**
Koshimura et al. (2009) proposed fault parameters for the 2004 Indian Ocean tsunami case. The fault was divided into 6 segments where accumulative energy is similar to total energy generated by the fault. The following table shows the details of the fault proposed by Koshimura et al. (2009).

| Segment | H (km) | L (km) | W (km) | Strike (°) | Dip (°) | Slip (°) | Dislocation (m) |
|---------|--------|--------|--------|------------|---------|----------|-----------------|
| 1 | 10 | 200 | 150 | 323 | 15 | 90 | 14 |
| 2 | 10 | 125 | 150 | 335 | 15 | 90 | 12.6 |
| 3 | 10 | 180 | 150 | 340 | 15 | 90 | 15.1 |
| 4 | 10 | 145 | 150 | 340 | 15 | 90 | 7.0 |
| 5 | 10 | 125 | 150 | 345 | 15 | 90 | 7.0 |
| 6 | 10 | 380 | 150 | 7 | 15 | 90 | 7.0 |

Detail of the location of the segments can be seen in the following figure.

[Figure]

Figure 1. Location of the six segments of the faults proposed by Koshimura et al. (2009).

The result of this multi-fault has been validated at the onshore area of Banda Aceh using measured flow-depths and flow-heigths. More complete explanation of this can be seen in Koshimura et al. (2009). We decide not to include the table and the figure to allow readers to read a more complete and rigour studies done by Koshimura et al. as briefly explained here.

For 8.5 Mw, we follow suggestions made by Horspool et al. (2014 ).
We based our simulations on the parameters with strike of $329^o$, dip $8.0^o$, slip $110^o$, and depth of 10 km. The 8.5 Mw simulation use single fault scenario where the location of the fault has been moved along the fault lines to obtain maximum impacts on Banda Aceh coast. We agree with the referee to add the explanation of the 8.5 Mw simulation fault scenario in or revised manuscript. Please see section 3.3 Earthquake scenarios in revised manuscript.

**COMMENT 2:**
Figure 10 and Figure 11, caption, correct to ". . .with BORR (right)"

**RESPONSE 2:**
Thank you for the detail correction. We will revise the caption on Figure 10 and Figure 11 .....
"without BORR (right)" with "with BORR (right)"

**COMMENT 3:**
Table 1. In COMCOT, the Manning roughness coefficents will not function when the SWE type is "Linear", so the second last column should be set to "None" when the SWE type is "linear"

**RESPONSE 3:**
Thank you, we will replace Manning Roughness Coefficients "0.02" to "None" as suggested. The revised table can be seen as follows (Table 1).

| Layer | Latitude | Longitude | Number of Grid | Ratio | Grid size (m) | Time Step (sec.) | Manning Roughness Coefficients | SWE type |
|---|---|---|---|---|---|---|---|---|
| 1 | 0.1 / 14.93 | 88.1 / 102.8 | 1772 | 1 | 1856 | 0.2 | None | Linear |
| 2 | 3 / 10 | 91 / 100 | 1920 | 2 | 928 | 0.1 | None | Linear |
| 3 | 4.08 / 8.98 | 92.05 / 97.98 | 3899 | 3 | 309.33 | 0.03 | None | Linear |
| 4 | 5.2708 / 6.695 | 94.51 / 95.99 | 3137 | 3 | 103.11 | 0.011 | None | Linear |
| 5 | 5.5 / 5.69 | 95.14 / 95.39 | 1426 | 3 | 34.37 | 0.004 | None | Linear |
| 6 | 5.515 | 95.235 | 2362 | 3 | 11.5 | 0.001 | Varied Coefficients (see **Table 3**) | Nonlinear |

---

## Author Comment (AC2) · 18 Dec 2018

First of all, we thank to comments from Referee 1 on our paper. The comments are very useful and we have adopted most of them in our revised manuscript submitted later. Now, allow us to deliver our responses to the comments of the Referee with respect and gratitude. We put word COMMENT# to represent the comments from the referee and the word RESPONSE # as our response to the comment. We also attach one Supplement document containing figures, equations, and a table related to our main responses.

GENERAL COMMENTS FROM REFEREE 1: This paper discussed the effectiveness

of the elevated road as the construction for tsunami disaster mitigation in Banda Aceh by using numerical simulation. The elevated road was planned after 2004 Indian Ocean tsunami. However, the detailed evaluation of the road as the disaster prevention facility has not been conducted. In this paper, 4 scenarios were prepared for the tsunami inundation simulation. These scenarios included the change of land use in the city of Banda Aceh. Banda Aceh has been recovering and developing after 2004 tsunami disaster. Therefore, the viewpoint of land use is important to discuss the effect of the disaster prevention facility, such as the elevated road, in near future. The scenerario of the magnitude of earhquake is quite severe, 8.5 and 9.15 Mw, and the reduction of the tsunami inundation area by the elevated road is not sufficient. But the effect of the road was confirmed and it is inferer that the road will have some effective functions for the disaster mitigation against the tsunami generated by earthquake smaller than 2004 Indian Ocean Tsunami. Conse- quently, it is expected that the results of this paper lead to the more detailed plannning of the elevated road and discussition as the disaster prevention facility. Then, this paper will contribute to the future tsunami disaster mitigation and development of Banda Aceh city.

OUR RESPONSE: Thank you very much for the referee 1's comments. We highly appreciate the comments and adopt them in our revised manuscript. This paper was inspired by lack of structural mitigation for tsunami hazards in Banda Aceh despite massive impacts found due to the 2004 Indian Ocean tsunami. One of the reasons for this situation is due to weak of financial capacity of the city to construct buildings, sea walls, or other types of tsunami mitigation structures. On the other hand, the tsunami is still an eminent threat to the city. Therefore, to seek alternatives for tsunami structural mitigation, we attempt to adopt concept of co-benefits structure. The co-benefits structure is a concept to put a structure or infrastructure to facilitate other functions than its main function. In this study, we select a road that is planned to develop along the coastal area of the city which will be mainly aimed at reducing traffic congestions in the city. In this study, we propose the road to also be functioned for tsunami wave reduction by modifying its designed. The road was initially planned to

be constructed at almost similar level to its original soil surface. Here, we propose to elevate the road about 5 m above the mean sea level (embankment road type). The similar type of the road was proven effective to stop tsunami wave on-shore propagation as in the case of Tobu Highway during the 2011 Great East Japan Earthquake and Tsunami in Sendai.

COMMENT 1: In Abstract: The condition of BORR should be written briefly.

RESPONSE 1: We have added the following statements to explain about the BORR. "The road will transect several lagoon, settlements, and bare land around the coast of Banda Aceh. Beside its main function to reduce traffic congestion in the city, the BORR is also proposed to reduce impacts of future tsunamis."

COMMENT 2: p.1, L.38-39: The relationship between 0.5m and "10% higher" is un-known.

RESPONSE 2: The study done by Horspool et al. published in 2014 was based on probabilistic tsunami hazards analysis where some recorded tsunami were used to calculate probable tsunamis in this area. For the area around Banda Aceh, the study revealed that tsunami wave height as high as 0.5 m could impact the region about 10% higher than lower tsunami wave height. This is one of the reasons to introduce the elevated road as a structure that could reduce tsunami energy for smaller intensity tsunami than the 2004 Indian Ocean tsunami.

COMMENT 3: p.3, L.2: "about 500m from its coastal line from any settlement". Where is it "500m" from?

RESPONSE 3: An area with a-500 m of width was intended to be deserted from any settlement. The 500 m was measured perpendicular from the coastline of the city.

COMMENT 4: p.3, L.15-16: From Fig.3, is the road on the revetment?

RESPONSE 4: No, the picture in Figure 3 was intended to show the existing condition around the coastal area of Banda Aceh. The road will be constructed along a transect

as can be seen in Fig. 1.

COMMENT 5: p.3, L.23: Has the problem (increase of land price) been solved already? Is there a possibility of restarting the project?

RESPONSE 5: A preliminary feasibility study by the municipality office has been performed to assess the land price and suitability. The study concluded that the road will attempt to avoid to cross settlement area whenever possible as this could create other problems.

COMMENT 6: p.4, L.13,14: In linear SWE, P=hu and Q=hv are correct. However, are these correct in nonlinear SWE?

RESPONSE 6: Yet, it is still applicable for nonlinear SWE mode.

COMMENT 7: p.4, L.33-34: Where is the reference height for "3m". If it is from the mean sea level, the heigt of the elevated road is less than 3.0m. But if it is from the original ground level, the height is 3.0m. At p.3, L.26, the term of "initial ground" is used.

RESPONSE 7: Thank you for looking at this mistake. It shoud be plus 5 m from mean sea level. We have revised it accordingly.

COMMENT 8: p.4, L.33-34: How much is the width of the road? Is the grid size of 11.5m in Layer 5 enough for the description of the road?

RESPONSE 8: The width of the planned road is 30 m. As our grid size in this layer was 11.5 m, we use three grids to represent the width of the road (BORR). This is sufficient to mimic the road widht in our models.

COMMENT 9: p.5, L.12-13: The land use in 2004 and its plan for 2029 should be shown. And the difference should be explained.

RESPONSE 9: Graphical differences between the 2004 land use and the 2029 land use can be seen in Figure 6 and Figure 7. Here we can see that the differences are

largely found in the ponds area and coastal forest area. The ponds area, where mainly located in the coastal lagoon, will be decreased. Meanwhile, coastal forest area will be increased.

COMMENT 10: p.6, L.19-20: How did you calculate the percentage of 1.2%? If this value is "% of total decrease" in Tab.5, what is the meaning of this value? If you want to say the effect of the land use change from 2004 to 2029 with BORR, you should calculate this value using 1252.0 (2004 with BORR) and 1203.47 (2029 with BORR). It is about 3.9% (>1.2%).

RESPONSE 10: The percentage here means the ratio of total tsunami inundation area to the area of Banda Aceh. Percentage of the decrease was calculated as follows. (Percentage of tsunami inundation area based on land use 2029 and magnitude 8.5 mw) – (percentage of tsunami inundation area based on land use 2004 and magnitude 8.5 mw) = 22.51%-21.33% = 1.18% or about 1.2%. The percentage of the total decrease means that the difference of total inundation are after BORR constructed. For example: for scenario of 8.5 mw without BORR (simulation #111) the inundation area was = 1.591,73 ha and for scenario 8.5 Mw with BORR (simulation #112 ) the inundation area was 1.252,20 ha. Therefore, the tsunami inundation area was decreased 339.53 ha, (1.591,73 ha - 1.252,20 ha = 339.53 ha ). Hence, the total decrease was = (339.53/1591,73) x 100% = 21.33%. In the section, we explained the total of tsunami inundation area decreased using scenario of land use 2029. Based on these explanation, we confirm that the statements are correct.

COMMENT 11: p.6, L.24: What is the reason of "Interestingly"?

RESPONSE 11: The elevated road which was not intended to reduce tsunami waves. During the 2011 Great East Japan Earthquake and Tsunami, the road was the limit of the tsunami inundation area. This proved that the structure could also be functioned as a secondary measure to reduce impacts of tsunami waves. Based on these, we acknowledge the provided facts are interesting and worth to be adopted as co-benefit

structures for tsunami mitigation.

COMMENT 12 : p.6, L.29-30: Where are the bridges in Fig.12?

RESPONSE 12: The bridges are located across the Flood Way, Krueng Aceh River, and Krueng Neng river. In Figure 12, the cross sections did not pass the bridge.Therefore, the bridge can not be seen. We have added the bridge positions in Figure 1. Please see the supplement part of this response (Fig. 1 in the supplement) to confirm the revision has been made.

COMMENT 13: p.6, L.31, "travel about 6km along the main rivers.": It is impossible to confirm the 6km-inundation in Fig.12.

RESPONSE 13: Yes, Fig. 12 could not explain the length of the inundation area along the main rivers. However, Fig. 10 could better explain our arguments. Cross-sections illustrated in Fig. 12 were intended to demonstrate the decrease of tsunami depths before and after BORR with various scenarios.

COMMENT 14: p.6, L.31: What is the reason of "higher than 1.5m"?

RESPONSE 14: This statement was intended to explain the condition of the river embankment in the city that was constructed higher than 1.5 m from the original soil surface. About 20 years ago, the embankment levels were increased to about plus 3.0 m from the soil through a series of construction projects.

COMMENT 15 : p.6, L.36-37: There is no discussion about velocity before. Is it possible to mention the effect of velocity reduction by the elevated road here?

RESPONSE 15 : Thank you for your suggestion. The reduction of tsunami velocity due to obstacles, naturally and man-made structures, have been proven correct by previous researches (e.g. Nandasena et al., 2012; Matsutomi and Okamoto, 2010). Sea walls as well as other types on-shore structures will reduce energy of the tsunami mainly by reducing the wave's velocity. Froude numbers will be reduced as the tsunami hit natural barriers or other solid man-made structures. Similar explanations have been added in

Section 4.2 to initiate explanation impacts of the elevated road. This will appear in our revised manuscript.

COMMENT 16: p.7, L.9: It is hard to recognize this value (difference) in Fig.11. You should mention Tab.5 for this difference.

RESPONSE 16: We have added a statement to refer to Table 4 as suggested.

COMMENT 17: p.7, L.11, "...about 5m with...": From the figure, "4m" is proper

RESPONSE 17: Thank you for suggesting the heights reduction to 4 m. We accept the suggestion and has revised it in our manuscript, accordingly.

COMMENT 18: p.7, L.18: What kind of "damages" do you considered? There is no explanation about the concrete type of damage.

RESPONSE 18: The damages caused by the tsunami has been studied by Suppasri et al. (2011) and Suppasri et al. (2012b). We have added the two references to specifically refer to the types of damages meant in this study.

COMMENT 19: p.7, L.18: How did you calculate "about 22%"?

RESPONSE 19: This is an average value for the a tsunami generated by an-8.5 Mw. Please see Table 5 column 5 for the comparison of the case. We have moved the sentence to suit the context of the percentage before discussing the 9.15 Mw case.

COMMENT 20: p.7, L.29: What is the meaning of "dynamic variable"?

RESPONSE 20: Land use changes from time to time. It follows regulations and people/economic demands on the other hand. In the case of Aceh, the land use planning is aimed to be implemented until 2029. It is important to note , despite the plan, it is possible some mid-term evaluation will be done and later also will change the land use of the city. Therefore, the land use could be regarded as dynamic variable in this study.

COMMENT 21: p.7, L.41: From Fig.12, tsunami wave with 3m height does not overflow

Interactive
comment

the structure. The content of this sentence is not consistent to the simulation results

RESPONSE 21: Thank you for suggesting us. The road will not be overflowed if the tsunami height is 3 m in front of the road at Transects A and C. However, it is important to see that at Transect B, the tsunami will overflow the structure and create a lower tsunami depth, which is between 0.8-1.5 m. Please refer to Figure 12.

COMMENT 22: p.8, L.3: What is the relation between the elevated road and the drainage system?

RESPONSE 22: We need to rigorously consider to re-design the city drainage system if the elevated road implemented. The elevated road could hamper surface runoff during rainy period. This could make flood problem in the city become worse. Therefore, it is important to consider to modify the city's drainage system if the elevated road adopted.

COMMENT 23: p.8, L.4-5: Is this sentence consistent to "p.3, L.26" and Fig.12

RESPONSE 23: The elevation is planned to be at +5.0 m from mean sea level. As the average soil surface is about 1.5 m, therefore, we add the words "or around 3.5 m from initial soil surface". To avoid confusing meaning, we delete the words "or around 3.5 m from initial soil surface".

COMMENT 24: p.8, L.19: What is the "co-benefits" for tsunami wave?

RESPONSE 24: A co-benefits structure is a concept to put a structure or infrastructure to facilitate other functions than its main function. In this study, we select a road that is planned to develop along the coastal area of the city which will be mainly aimed at reducing traffic congestions in the city. In this study, we propose the road to also be functioned for tsunami wave reduction by modifying its designed. The concept of co-benefit structure has been also studied in Sri Lanka (Samarasekara et al. , 2017). Similar explanation has been also provided at P.7 Line 43.

COMMENT 25: p.8, L.19: What is "Tsunami multidefense system"? There is no explanation.

RESPONSE 25: Tsunami multi-defense is a set of structures to mitigate impacts of tsunami. The concept was introduced in Tohoku region of Japan during rehabilitation and reconstruction process following the 2011 tsunami. The structures consist of sea-wall, coastal forests, canal that is parallel to coastline, escape hills, and eleavated roads.The multi-defense system has been studied by Koshimura et al. (2014) and Pakoksung et al. (2018). A similar explanation has been added in Page 2. Ilustration of the multi-layered system can be seen in Fig. 2 in the supplement part of this response.

Technical Corrections

COMMENT 26: p.4: In equation, "sin" and "cos" should be written in Roman style.

RESPONSE 26: Thank you, we have revise them in Roman style as suggested.

COMMENT 27: p.4: "Φ" should be changed to "ÏȚ" in eqs.(4)-(6).

RESPONSE 27 : In our manuscript we already put the symbol as "ÏȚ". Please confirm our response to the equation attached (see supplement part of our response).

COMMENT 28: p.4, L.16: "g" should be written in Italic style.

RESPONSE 28: Thank you, we will replace to Italic style for "g".

COMMENT 29: p.5, Eqs.(11) and (13): Are these descriptions correct? Is "logK" a variable, that is, is "log" not a function? Eq.(12) is the same.

RESPONSE 29: It means logarithmic values of K. They are correct.

COMMENT 30: p.6, L.21: ... about 25% both in 2004 and 2029.

RESPONSE 30: Yes, they are correct. Both similar percentages were found in 2004 and 2029 land use cases.

COMMENT 31:p.6, L21: "Figure 12" should be labeled as "Figure 10". Because this figure is refered before original Figure 10 and 11

RESPONSE 31: In Page 6 L.15, we have mentioned Figure 9. Figures 9,10 and 11

were shown earlier to describe the overall tsunami flow depths for each scenario. Later, analysis was done following three transects. Therefore, we put Figure 12 at the end of the manuscript.

COMMENT 32: p.6, L.22: ... the three transects indicated in Fig.1 that ...

RESPONSE 32: Thank you very much. We agree with the Refere. Therefore, we have modified it accordingly.

COMMENT 33: p.6, L.29, "could be reduced": "stopped" may be suitable in this case than "reduced".

RESPONSE 33: In transect B, the tsunami wave could not be stopped. It will overflow the elevated road but will create shorter distance of tsunami run-up. Therefore, we would like to maintain the word "reduced" at this sentence.

COMMENT 34: p.6, L.41: The number of figures should be changed by change of figure number of Fig.12.

RESPONSE 34: Our response to this is similar to COMMENT No. 31.

COMMENT 35: p.7, L.35: "Tokida and Tanimoto, 2014" is not found in the references.

RESPONSE 35: The reference is already in the reference list. Please find the following: Tokida, K. and Tanimoto, R.: Lessons for countermeasures using earth structures against tsunami obtained in the 2011 Off the Pacific Coast of Tohoku Earthquake, Soils and Foundations, 54(4), 523-543, 2014.

COMMENT 36: p.8, L.7, "overlapping process": "overtopping" or "overflowing" ?

RESPONSE 36: Thank you for pointing out this word. It is incorrect to use overlapping in this case. We have changed it to overflowing as this is a long-wave process.

COMMENT 37: Figure 4,5,9,10,11: These figures should be bigger.

RESPONSE 37: Thank you. We will show the larger figures.

COMMENT 38: Figure 10 (caption): The simulation were demonstrated by using land use ...

RESPONSE 38: We will revise the sentence with "The simulation were demonstrated by using land use type before the 2004 Indian Ocean Tsunami"

COMMENT 39: Figure 11 (caption): ...and with BORR (right).

RESPONSE 39: Thank you for the detail correction. We will replace "without" with "with"

COMMENT 40: Figure 12: What is "Elevated Road ($\pm$5.0m)"? In p.4, L.34, "to plus 3.0m from the mean sea level"

RESPONSE 40: We will revise the caption of elevated road in Figure 12 to "Elevated road $\pm$5.0 above MSL"

COMMENT 41: Figure 12: It is difficult to distinguish the difference of lines, especially yellow lines are unclear.

RESPONSE 41: Thank you. We will replace it with a more contrast color and clearer legends. Please refer to the Supplement part of this respons (see Fig. 2) for one of examples of the revised figures.

COMMENT 42: Figure 12 (legend): Simulation code should be written in the legend because the code is used in the main sentense.

RESPONSE 42: Thank you very much. In Table 2, We have determined to classify the simulation code and also in the main sentence. We agree with the Referee to use the simulation code in the legend of Figure 12. Therefore, we have revised it accordingly. Please see the Supplement part of this response (Fig. 2) as our confirmation to the revision.

COMMENT 43: Table 1 (title): ...setup of the six layers for ...

RESPONSE 43: Thank you, We will change "five layer" to "six layers"

COMMENT 44: Table 1: What is the "Ratio" in 5th column? If this is grid size ratio from parent layer to child layer, a blank is better in Layer 1.

RESPONSE 44: Yes, it is a grid size ratio from parent layer to child layer. Thank you, We will delete the ratio in Layer 1, make it a blank column.

COMMENT 45: Table 1 (Layer 6): Two values in Latitude and Longitude may indicate the locations of "start" and "end", respectively. But why is only one value in Layer 6?

RESPONSE 45: Thank you for your correction. We missed the bottom row. Layer 6 should have two values like the other ones. Thank you for the correction. We have modified the table as can be seen in the Supplement part of our response. The revised table has also adopted suggestions from Referee 2.

COMMENT 46: Table 5: What is "Ha"? Is this "ha" (hectare)?

RESPONSE 46: Yes, We mean hectare. We agree with the Refereee and will change "Ha" to "ha"

COMMENT 47: Table 5: Font size in the bottom row is slightly bigger than others.

RESPONSE 47: Thank you very much. We have corrected the font size and it will visible in our revised manuscript.

Please also note the supplement to this comment:
https://www.nat-hazards-earth-syst-sci-discuss.net/nhess-2018-276/nhess-2018-276-AC2-supplement.pdf

**Supplement:**

**Supplement Section**

Response to Referee 1: Assessing the tsunami mitigation effectiveness of a planned Banda Aceh Outer Ring Road (BORR), Indonesia

by Syamsidik et al.

[Figure]

Fig. 1 The study area (revised).

[Figure]

Figure 2. The Multi-layered tsunami defense as depicted by Koshimura et al. (2014)

The equations 4 and 6 that confirm the  use of $\phi$

$$\frac{\partial \eta}{\partial t} + \frac{1}{R\cos\phi}\left\{\frac{\partial P}{\partial \psi} + \frac{\partial}{\partial \phi}(\cos\phi Q)\right\} = -\frac{\partial h}{\partial t}, \tag{4}$$

$$\frac{\partial Q}{\partial t} + \frac{1}{R\cos\phi}\frac{\partial}{\partial \psi}\left\{\frac{PQ}{H}\right\} + \frac{1}{R}\frac{\partial}{\partial \phi}\left\{\frac{Q^2}{H}\right\} + \frac{gH}{R}\frac{\partial \eta}{\partial \phi} + fP + F_y = 0, \tag{6}$$

[Figure]

Fig. 3 an example of revised Fig. 12 in the manuscript to show more visible lines and legends. Similar revisions have been done to Transect B and Transec C.

Table 1. Information on the Setup of the six layers for COMCOT Simulations

| Layer | Latitude | Longitude | Number of Grid | Ratio | Grid size (m) | Time Step (sec.) | Manning Roughness Coefficients | SWE type |
|---|---|---|---|---|---|---|---|---|
| 1 | 0.1 | 88.1 | 1772 | | 1856 | 0.1 | none | Linear |
| | 14.93 | 102.8 | | | | | | |
| 2 | 3 | 91 | 1920 | 2 | 928 | 0.05 | none | Linear |
| | 10 | 100 | | | | | | |
| 3 | 4.08 | 92.05 | 3899 | 3 | 309.33 | 0.017 | none | Linear |
| | 8.98 | 97.98 | | | | | | |
| 4 | 5.2708 | 94.51 | 3137 | 3 | 103.11 | 0.006 | none | Linear |
| | 6.695 | 95.99 | | | | | | |
| 5 | 5.5 | 95.14 | 1426 | 3 | 34.37 | 0.002 | none | Linear |
| | 5.69 | 95.39 | | | | | | |
| 6 | 5.515 | 95.235 | 2362 | 3 | 11.5 | 0.001 | Variable Manning Roughness Coefficients (see **Table 3**) | Nonlinear |
| | 5.615 | 95.378 | | | | | | |